# *Homo naledi* and Pleistocene hominin evolution in subequatorial Africa

Lee R Berger[1]*, John Hawks[1,2], Paul HGM Dirks[1,3], Marina Elliott[1], Eric M Roberts[1,3]

[1]Evolutionary Studies Institute, University of the Witwatersrand, Johannesburg, South Africa; [2]Department of Anthropology, University of Wisconsin, Madison, United States; [3]Department of Geosciences, James Cook University, Townsville, Australia

**Abstract** New discoveries and dating of fossil remains from the Rising Star cave system, Cradle of Humankind, South Africa, have strong implications for our understanding of Pleistocene human evolution in Africa. Direct dating of *Homo naledi* fossils from the Dinaledi Chamber (*Berger et al., 2015*) shows that they were deposited between about 236 ka and 335 ka (*Dirks et al., 2017*), placing *H. naledi* in the later Middle Pleistocene. Hawks and colleagues (*Hawks et al., 2017*) report the discovery of a second chamber within the Rising Star system (*Dirks et al., 2015*) that contains *H. naledi* remains. Previously, only large-brained modern humans or their close relatives had been demonstrated to exist at this late time in Africa, but the fossil evidence for any hominins in subequatorial Africa was very sparse. It is now evident that a diversity of hominin lineages existed in this region, with some divergent lineages contributing DNA to living humans and at least *H. naledi* representing a survivor from the earliest stages of diversification within *Homo*. The existence of a diverse array of hominins in subequatorial comports with our present knowledge of diversity across other savanna-adapted species, as well as with palaeoclimate and paleoenvironmental data. *H. naledi* casts the fossil and archaeological records into a new light, as we cannot exclude that this lineage was responsible for the production of Acheulean or Middle Stone Age tool industries.

*For correspondence: Lee. Berger@wits.ac.za

Competing interests: The authors declare that no competing interests exist.

## Introduction

Recent work within the Rising Star cave system has given rise to two findings that influence our knowledge of *Homo naledi*, its behavior and its position in human evolution. The hominin deposit in the Dinaledi Chamber, which comprises the first described sample of *H. naledi* (*Berger et al., 2015*), represents individuals that probably lived between 236 ka and 335 ka (*Dirks et al., 2017*). A second chamber with a rich fossil deposit, known as the Lesedi Chamber, contains multiple individuals of *H. naledi* including a partial skeleton (*Hawks et al., 2017*). The Lesedi remains are morphologically very similar to those in the Dinaledi Chamber, consistent with the hypothesis that together these two chambers represent a single hominin population (*Hawks et al., 2017*). They occur in a depositional context similar to, but geologically separate from, the Dinaledi Chamber (*Hawks et al., 2017*).

Considered together, these discoveries change our interpretation of many aspects of the Pleistocene fossil and archaeological record. In this paper, we develop a hypothesis that places Pleistocene hominins into the broader context of the biogeography of Africa. In this vast region, finding a fossil hominin species like *H. naledi* that differs substantially from known fossil samples is not surprising in light of the biogeography of other mammals, the genetic record of humans, and the previously weak fossil record. Genetics and the fossil record together suggest that the ancestors of modern humans evolved in Africa amid a diversity of hominin populations and species, which shared many aspects of the human adaptive pattern.

**eLife digest** Species of ancient humans and the extinct relatives of our ancestors are typically described from a limited number of fossils. However, this was not the case with *Homo naledi*. More than 1,500 fossils representing at least 15 individuals of this species were unearthed from the Rising Star cave system in South Africa between 2013 and 2014. Found deep underground in the Dinaledi Chamber, the *H. naledi* fossils are the largest collection of a single species of an ancient human-relative discovered in Africa.

After the discovery was reported, a number of questions still remained. *H. naledi* had an unusual mix of ancient and modern traits. For example, it had a small brain like the most ancient of human-relatives, yet its wrists looked much like those of a modern human. This raised the question: where does *H. naledi* fit within the scheme of human evolution?

Now, Berger et al.—who include many of the researchers who were involved in the discovery of *H. naledi*—reconsider this question in the light of new findings reported in two related studies. First, Dirks et al. provide a long-anticipated estimate for the age of the fossils at between 236,000 and 335,000 years old. Second, Hawks et al. report the discovery of more *H. naledi* fossils from a separate chamber in the same cave system.

These estimated dates fall in a period called the late Middle Pleistocene, and mean that *H. naledi* possibly lived at the same time, and in the same place, as modern humans. Berger et al. explain that the existence of a relatively primitive species like *H. naledi* living this recently in southern Africa is at odds with previous thinking about human evolution. Indeed, all other members of our family tree known from the same time had large brains and were generally much more evolved than our most ancient relatives. However, Berger et al. argue that we have only an incomplete picture of our evolutionary past, and suggest that old fossils might have been assigned to the wrong species or time period.

Reassessing the old fossils might lead the scientific community to rethink what kinds of human-relative were around in southern Africa at different times, and what those ancient species were capable of. For example, archeologists had previously thought that modern humans made all the stone tools dating from around the late Middle Pleistocene found in southern Africa, but now we must consider whether some of them could have been made by *H. naledi*.

## The Pleistocene fossil record and *H. naledi*

Geological evidence from the Dinaledi Chamber, including direct dating of the fossil hominin remains, places *H. naledi* in the later Middle Pleistocene. All considerations of the phylogenetic placement of *H. naledi* to date agree that its branch on the hominin phylogeny must have originated earlier than 900 ka (*Dembo et al., 2016*), and earlier branch points are credible (*Berger et al., 2015*; *Dembo et al., 2016*; *Thackeray, 2015*). If *H. naledi* stems from a basal node within *Homo*, a scenario that is not rejected by phylogenetic analyses (*Dembo et al., 2016*; *Hawks and Berger, 2016*), its branch may have originated much earlier, in the Pliocene. How did this *H. naledi* lineage fit into the hominin diversity of the Early and Middle Pleistocene, and is it possible that palaeoanthropologists may already have discovered—but not recognized—other fossils that represent this branch?

The hominin fossil record of the African Middle Pleistocene is extremely sparse (*Klein, 1973*; *Berger and Parkington, 1995*; *Grün et al., 1996*; *Stynder et al., 2001*; *Marean and Assefa, 2005*; *Klein et al., 2007a*; *Millard, 2008*; *Stringer, 2011*; *Wood, 2011*; *Smith et al., 2015*). Fossils that putatively derive from this period between 780,000 and 130,000 years ago are limited and typically fragmentary (*Table 1*; *Figure 1*). Only a small number are thought to come from the period before 200,000 years ago. To this number, it is possible to add perhaps a half dozen partial mandibles and a somewhat greater number of postcranial fragments or dental remains. Many of these were found prior to 1960 and lack adequate provenience. Some have been 'dated' mainly by their morphological appearance, or by examination of vertebrate faunal remains that were not excavated together in association with the hominin specimen. Others have been subject to direct dating, but this has often been reported in ways that do not reflect the full statistical uncertainty (*Millard, 2008*). Among these

**Table 1.** Significant hominin fossil remains from the Middle and Early Late Pleistocene of Africa. Included are those sites that have geological age estimates between 780,000 and 120,000 years ago, and some sites for which claims of Middle Pleistocene age have been made but without chronometric support. Sites denoted here with 'no date' are those for which no chronometric determinations based on samples of hominin material or securely associated faunal remains have been reported in the literature. Some chronometric determinations that were based only on morphology or associated fauna have given rise to broad age estimations; we omit the details of such determinations here. Some additional sites with fragmentary remains, especially isolated dental remains, are not listed. The first four entries (KNM-OL 45500, OH 12, Daka and Buia) are older than the beginning of the Middle Pleistocene but are included because they are discussed in text.

| Site | Specimens | Location | Geological age (ka) | Source(s) |
|---|---|---|---|---|
| Olorgesailie (KNM-OL 45500) | Frontal | Kenya | 900–970 | *Potts et al. (2004*) |
| Olduvai Gorge (OH 12)* | Partial calvaria | Tanzania | 780–1,200 | *Tamrat et al. (1995)*; *Mcbrearty and Brooks (2000)* |
| Daka | Calvaria, femur | Ethiopia | ~1,000 | *Asfaw et al. (2002)* |
| Buia | Calvaria, postcranial fragments | Eritrea | ~1,000 | *Abbate et al. (1998)* |
| Tighénif (Ternifine) | Three mandibles, skull fragment | Morocco | ~700 | *Geraads et al. (1986)* |
| Elandsfontein (Saldanha) | Partial calvaria and mandible frag | South Africa | 600–1,000 | *Klein et al. (2007a)* |
| Bodo | Partial calvaria, left parietal (found roughly 400 m from Bodo 1), distal humerus | Ethiopia | 550–640 | *Conroy et al. (1978)*; *Clark et al. (1994)* |
| Baringo (Kapthurin Formation) | Mandible, ulna | Kenya | 510–512 | *Leakey et al. (1970)*; *Deino and McBrearty (2002)* |
| Salé | Partial calvaria and upper dentition | Morocco | ~300 | *Jaeger, 1975*); *Geraads (2012)* |
| Ndutu* | Partial calvaria | Tanzania | 370–990 | *Tamrat et al. (1995)*; *Mcbrearty and Brooks (2000)* |
| Berg Aukas | Partial femur | Namibia | No date | *Grine et al. (1995)* |
| Kabwe | Calvaria, material from at least three individuals | Zambia | No date | *Klein (1973)* |
| Florisbad | Partial cranium | South Africa | 224–294 | *Grün et al. (1996)* |
| Cave of Hearths | Partial mandible | South Africa | No date | *Cooke, 1962*) |
| Hoedjiespunt | Teeth, tibia | South Africa | No date | *Berger and Parkington (1995)*; *Stynder et al. (2001)* |
| Eliye Springs | Calvaria | Kenya | No date | *Bräuer and Leakey (1986)* |
| Dinaledi Chamber (Rising Star) | Remains of at least 15 individuals | South Africa | 236–335 | *Berger et al., 2015*; *Dirks et al. (2017)* |
| Lesedi Chamber (Rising Star) | Partial skeleton, remains of at least three individuals | South Africa | No date | *Hawks et al., 2017*) |
| Omo Kibish | Two partial crania, partial skeleton | Ethiopia | 155–200 | *McDougall et al. (2005)*; *Aubert et al. (2012)* |
| Herto | Three partial crania | Ethiopia | 154–160 | *White et al. (2003)*; *Clark et al. (2003)* |
| Ileret (KNM-ER 3884) | Partial calvaria | Kenya | 162–∞ | *Bräuer et al. (1997)* |
| Jebel Irhoud | Three calvaria, mandible, fragments of seven individuals | Morocco | 144–176 | *Hublin (2001)*; *Smith et al. (2007)* |
| Laetoli (Ngaloba Beds) | Cranium | Tanzania | 130 | *Day et al. (1980)*; *Hay (1987)* |
| Singa | Calvaria | Sudan | 131–135 | *McDermott et al. (1996)* |
| Lake Eyasi | Calvaria | Tanzania | 88–132 | *Mehlman (1984, 1987)*; *Domínguez-Rodrigo et al., 2008* |

*Many authors have studied the stratigraphy of Olduvai Gorge and nearby sites, resulting in varied dates being reported for these fossils. We report here the widest range as reviewed by *Mcbrearty and Brooks (2000)*, based on the paleomagnetic sequence.

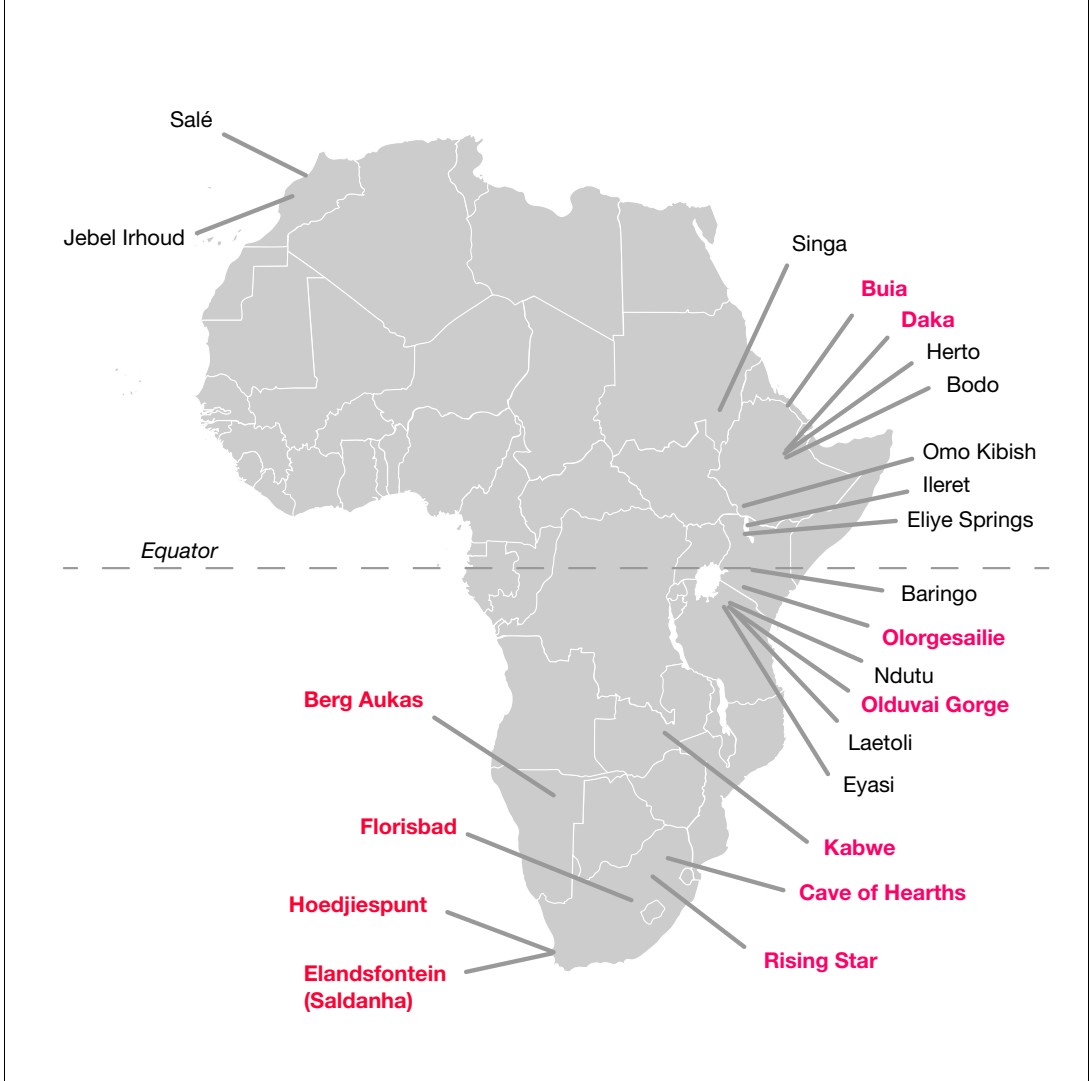

**Figure 1.** African fossil sites from the Middle and earliest Late Pleistocene. Sites discussed in the text are highlighted in pink here. Geological age estimations for each fossil hominin assemblage are given in *Table 1*, along with references.

finds, crania have been considered the most diagnostically important as indicators of the presence of archaic humans with large brain sizes in the Middle and Late Middle Pleistocene.

Only three of the fossil crania occur within 3,000 km of the location of the Rising Star cave system. The cranium and mandible fragment from Elandsfontein, or Saldanha, South Africa derives from an open-air deposit with multiple depositions of faunal remains, which, by comparison with East African vertebrate assemblages, come from about 600,000 to 1 million years ago (*Klein et al., 2007a*). However, the skull and mandible fragment are essentially surface finds. The Florisbad partial cranium from near Bloemfontein, South Africa, comes from mineral springs which accumulated fossils from around 40 ka to 400 ka; a direct electron spin resonance (ESR) assessment on a human $M^3$ yielded an age estimate of 259 ± 35 ka (*Grün et al., 1996*). This single tooth is assumed to be associated with the fossil hominin cranium. It is roughly 1 mm smaller in diameter than a sample of four *H. naledi* upper third molars, and lies within the size range of both *H. erectus* and African modern human populations (*Smith et al., 2015*). Although third molars vary substantially in humans and fossil *Homo*, this tooth does not resemble the morphology of known *H. naledi* maxillary third molars. The Kabwe, or Broken Hill, cranium from Zambia has no direct geological age assessment and its context is very poorly known. The vertebrate fauna in the cave sediments where the skull presumably

originated is Middle Pleistocene in age (*Klein, 1973*; *Millard, 2008*), although the context of the skull in relation to these faunal remains is not clear. Observing that the morphology of the skull is primitive compared to those of crania from early Late Pleistocene contexts, some workers have argued that the Kabwe skull may be 300 ka or earlier (*Bräuer, 2008*), though more recent work suggests it may be of later middle Pleistocene age (*Stringer, 2011*). The Lake Ndutu cranium is the only other subequatorial cranial specimen thought to be between 780,000 and 200,000 years old, but it also lacks secure provenience or a direct geological age estimate. Teeth from Hoedjiespunt (*Berger and Parkington, 1995*; *Stynder et al., 2001*) and the mandible from Cave of Hearths, South Africa, also provide evidence of hominins, probably of Middle Pleistocene age, but without further comparisons we cannot rule out the possibility that these fossils may themselves represent *H. naledi*. All of the cranial fossils discussed above share substantially larger brain size than *H. naledi* but are morphologically diverse in comparison to each other. The uncertain provenience and inexact geological ages of these fossils limit our ability to test when and where the populations that they represent may have existed, and it is conceivable that some of the remains are not Middle Pleistocene at all.

In addition to the crania, a few postcranial specimens document individuals with a larger body size than has yet been observed for *H. naledi*. The Kabwe hominin collection includes several postcranial elements, which are not associated with certainty with the cranium, but that clearly represent individuals with a large body size. Also within the 3,000 km radius are a large femur from Berg Aukas, Namibia (*Grine et al., 1995*), presumed to be of Middle Pleistocene age, and a large tibia from Hoedjiespunt, South Africa (*Berger and Parkington, 1995*; *Stynder et al., 2001*; *Churchill et al., 2000*) that is Middle Pleistocene in age.

Earlier than these Middle Pleistocene fossil specimens, the sites of Olorgesailie and Olduvai Gorge both preserve evidence of hominin crania (KNM-OL 45500, OH 12), which although fragmentary, clearly belonged to individuals that had relatively small brain sizes comparable to some of the earliest *H. erectus* remains (*Potts et al., 2004*; *Antón, 2004*). These have been called 'H. erectus-like', but they are different from contemporary fossil specimens attributed to *H. erectus* from further to the north, such as BOU-VP-2/66 calvaria from Daka, Ethiopia (*Asfaw et al., 2002*; *Gilbert et al., 2003*), and the UA 31 cranium from Buia, Eritrea (*Abbate et al., 1998*), both approximately 1 million years old. KNM-OL 45500 and OH 12 also differ from the large and robust OH 9 cranium, which is likewise from Olduvai Gorge but earlier in time. The fossil sites from Lake Baringo southward to Olduvai Gorge lie at the hinge point separating subequatorial from northeast African populations of many living species of mammals. It is possible that fossils such as KNM-OL 45500, OH 12 and OH 28 represent northern excursions of a more diverse subequatorial hominin community that included *H. naledi* and its relatives.

## Biogeography of subequatorial Africa

The great tropical forests of Africa pose a biogeographic barrier to species that are adapted to savanna and savanna-woodland-mosaic habitats. During the Pleistocene, these tropical forests repeatedly expanded eastward along the equator. During these times, Equatorial East Africa broke into a mosaic of small savanna remnants, while a large and contiguous area of savanna and savanna-woodland mosaic stretched southward from the equatorial forest (*Lorenzen et al., 2012*; *Faith et al., 2016*). During most of the Pleistocene, the area suitable for hominins located to the south of the equatorial forests was vastly larger than that to the north or east. During the last 1.3 million years of this period, the Lake Malawi basin became increasingly moist with greater climate stability (*Johnson et al., 2016*). The paleoclimate record for the Kalahari is not as deep, but during the last 200,000 years, this area underwent both arid periods with dune formation and wetter periods with vast paleolakes (*Robbins et al., 2016*), a pattern that likely held during earlier climate cycles. Hence the subequatorial area that was potentially suitable for hominins varied extensively, but may have been 5–15 times larger than the equivalent habitats in eastern Africa north of the equator.

This biogeographic history is reflected in the phylogeography of savanna-adapted species today. Across several orders of savanna-adapted mammals, including ungulates, primates, and carnivores, a north–south dichotomy can be observed in genetic patterning (*Bertola et al., 2016*), and this is thought to be the result of past climate patterns. A comparison of 19 ungulate species shows that many have a 'suture zone' in the East African equatorial region, where relatively distinct mtDNA clades from eastern and southern parts of the species' range meet (*Lorenzen et al., 2012*). In several species of ungulates, primates, and carnivores, today's East African populations were colonized

from the large and more contiguous region to the south; in some other species, East African diversity has been maintained through vicariance of populations in mosaic refugia within East Africa. A southern African origin is also inferred for baboons (*Papio*). Likemany savanna ungulates this genus manifests an mtDNA suture zone in equatorial East Africa, which has recently been expanded by introgressive hybridization of a more southern mtDNA clade into *P. cynocephalus* (*Zinner et al., 2013*).

Similar to many other mammals in Africa, living humans carry a genetic legacy of greater subequatorial African diversity. Today, the human populations that have the highest observed genetic diversity are the hunter-gatherers of southwestern Africa, followed by the Mbuti rainforest hunter-gatherers of north eastern Congo (*Mallick et al., 2016*). Hadza and Sandawe people from Tanzania also draw ancestry from a diverse source population similar to today's southwestern African hunter-gatherers (*Pickrell et al., 2014*). Today's populations are descendants of human populations that have been relatively large and stable across most of the past 200,000 years. These populations did not undergo the population bottleneck evidenced in non-African populations and to a lesser degree in West African and northeastern African peoples (*Mallick et al., 2016*). Additionally, the genomes of Hadza, Sandawe, Biaka, Baka, and San people bear evidence of a small fraction of introgression from highly genetically divergent populations that no longer exist (*Hammer et al., 2011*; *Lachance et al., 2012*; *Beltrame et al., 2016*; *Hsieh et al., 2016*). The implication of these observations is that tropical and subequatorial Africa were home to multiple genetically divergent populations of hominins. Some of these populations diverged in the Early Pleistocene, and had genomes that were equally or more diverse than those of Neanderthals, Denisovans, or contemporary modern humans. Some of these populations survived and hybridized after the initial diversification of modern humans, perhaps as recently as 35,000 years ago (*Hammer et al., 2011*) or even into the early Holocene (*Hsieh et al., 2016*). As other have noted (*Stringer, 2016*), the fossil hominin record of the Middle and Late Pleistocene shows no simple linear progression towards modern humans, and different morphological forms overlapped in time. A small-brained hominin has been recognized from this time period in Asia on the island of Flores (*Brown et al., 2004*), and we now include a small-brained species of hominin from Africa in this recognized diversity.

## Modern humans are a relict species

Modern *H. sapiens* is a phylogenetic relict. In biology, a relict is a species that remains from a clade that was more diverse in the past (*Grandcolas and Trewick, 2016*). We have known for a long time that other hominin populations once inhabited Eurasia and island Southeast Asia, including the Neanderthals, Denisovans, and *H. floresiensis* (*Bocquet-Appel and Demars, 2000*; *Brown et al., 2004*; *Cooper and Stringer, 2013*; *Li et al., 2017*). Genetic evidence shows that equally diverse populations of archaic humans once existed in subequatorial Africa (*Hammer et al., 2011*; *Stringer, 2011*; *Lachance et al., 2012*), and although no fossil evidence can yet be associated with such evidence of genetic introgression, the Middle Pleistocene record of this region does speak to the presence of morphological diversity. Within this context, *H. naledi* provides fossil evidence of one subequatorial lineage, and we do not yet know whether it contributed to the modern human gene pool.

Another implication of modern humans as a relict is that the features of today's humans give a biased and incomplete picture of the diversity of the *Homo* clade (cf. *Grandcolas et al., 2014*). These biases have had enormous consequences for the historical development of paleoanthropology. One of the most persistent biases has been to conceive of postcranial and dental adaptations of *Homo* as mere adjuncts to the extraordinary increase in brain size evidenced in living humans. Poor fossil evidence once appeared to support the notion that human-like aspects of locomotor, manipulatory, and dietary strategy evolved in tandem with larger brains, and that *H. erectus* combined these for the first time (*Wood and Collard, 1999*; *Hawks et al., 2000*). But newer evidence shows that some fossils attributed to *H. erectus* had a mosaic of human-like and primitive postcranial features (*Lordkipanidze et al., 2007*), that some fossil samples of *H. erectus* had brain sizes equivalent to those of *H. habilis* and *H. rudolfensis* (*Lordkipanidze et al., 2013*), and that *H. habilis* may be closer to *Au. sediba* than to other species of *Homo* (*Dembo et al., 2015*, *2016*). *H. naledi* shows that many human-like anatomical aspects of the hand, foot, lower limb, dentition and cranium, including some aspects that are not present in *H. erectus*, occurred in a species with a brain size equal to that of australopiths (*Berger et al., 2015*).

The full geographic extent of *H. naledi* is unknown, though aspects of its anatomy might be used to argue that this species is unlikely to be endemic only to the region where its fossils are presently found. With its human-like pattern of lower limb and foot anatomy (*Harcourt-Smith et al., 2015*; *Marchi et al., 2017*), human-like stature (*Berger et al., 2015*), and dental morphology consistent with reliance on a high-quality diet (*Berger et al., 2015*), *H. naledi* appears to have used its environment in a similar way to *H. erectus* and *H. sapiens*. Indeed, even earlier small-bodied *Australopithecus* and *Paranthropus* have been inferred to be eurytopic, capable of using a wide range of habitats, and all well-sampled species are geographically widespread (*Wood and Strait, 2004*; *Behrensmeyer and Reed, 2013*). Aside from these considerations, the hominin fossil sample is insufficient to support further conclusions about the geographic range of *H. naledi*.

## What explains the mosaic anatomy of *H. naledi*?

The late survival of *H. naledi* from origins deep in the Pleistocene up to the Dinaledi and Lesedi Chamber deposits is surprisingly unhelpful in testing hypotheses about its evolutionary origin or its morphological pattern. How the traits of *H. naledi* evolved does not depend on the geological age of the Dinaledi Chamber fossils, but on the phylogenetic position of *H. naledi* and the morphological patterns of other hominin taxa.

Phylogenetic scenarios for *H. naledi* place its origin either: (1) somewhere among the poorly resolved branches leading to *H. habilis*, *H. rudolfensis*, *H. floresiensis* and *Au. sediba* (*Berger et al., 2015*; *Dembo et al., 2016*); *Thackeray, 2015*): (2) as a sister to *H. erectus* and larger-brained *Homo* including *H. sapiens* (*Dembo et al., 2016*); or (3) as a sister to a clade including *H. sapiens*, *H. antecessor*, and other archaic humans (*Dembo et al., 2016*) (*Figure 2*). Maximum parsimony analysis of a large dataset of cranial and dental traits supports scenario 1, placing *H. naledi* among the most basal nodes of the *Homo* phylogeny (*Dembo et al., 2016*). Bayesian analysis of the same dataset supports scenario 3, placing *H. naledi* closer to modern humans than any *H. erectus* sample (*Dembo et al., 2016*). An informal consideration of postcranial traits suggests that *Dembo et al. (2016)* analysis, if it included postcrania, might more likely support scenario 2. This is because *H. naledi* shares many derived features of the hand, foot, and lower limb with *H. erectus* and *H. sapiens* that are apparently absent from *H. habilis*, *H. floresiensis*, or *Au. sediba*, yet lacks several derived traits of the shoulder, trunk, and hip shared by *H. erectus* and *H. sapiens* (*Hawks et al., 2017*; *Williams et al., 2017*; *Marchi et al., 2017*; *Feuerriegel et al., 2017*). However, the fossil record for these areas of anatomy in early hominins other than *H. naledi* is admittedly limited.

No interpretation of this anatomy can eliminate the necessity of some reversals or parallelism. If *H. naledi* is a sister to *H. sapiens* (scenario 3), then all of the primitive traits it does not share with *H. erectus*, including its small brain size (*Berger et al., 2015*), shoulder morphology (*Feuerriegel et al., 2017*), ilium form (VanSickle et al., personal communication), long, anteroposteriorly flattened femur neck (*Marchi et al., 2017*), thorax shape (*Williams et al., 2017*), and markedly curved finger bones (*Kivell et al., 2015*), might be interpreted as evolutionary reversals. If *H. naledi* is a sister taxon to a clade including *H. habilis*, *H. rudolfensis* and all other large-brained species of *Homo*, then the larger brain size of these other species of *Homo* could be homologous. But this scenario would require many parallel evolutionary developments in *H. naledi* and *H. sapiens*, including hand and wrist morphology (*Kivell et al., 2015*), foot morphology (*Harcourt-Smith et al., 2015*), lower limb morphology (*Marchi et al., 2017*), and some cranial and dental morphologies (*Laird et al., 2017*; *Schroeder et al., 2017*). Some of these derived traits are also shared with *H. erectus*, others are not evidenced in any known *H. erectus* fossils. Whatever phylogenetic scenario we accept, *H. naledi* is not unique in demonstrating homoplasy (*Wood and Harrison, 2011*), but it does present a uniquely strong postcranial record documenting its mosaic anatomy.

The long evolutionary branch leading to *H. naledi* as represented in the Rising Star cave system may have implications for its mosaicism, at least with respect to cranial and mandibular form. Much of the evolution of cranial form among species of *Homo* in the Pleistocene appears to be consistent with neutral evolution by genetic drift, with a few features showing evidence of adaptive evolution (*Ackermann and Cheverud, 2004*; *Weaver et al., 2007*; *Schroeder et al., 2014*). If the correlations among some aspects of *H. naledi* cranial anatomy were not constrained by selection, then a long evolutionary branch would create substantial opportunity for divergence over time by drift. Such non-adaptive evolution, combined with the adaptive evolution of some traits, might create a unique

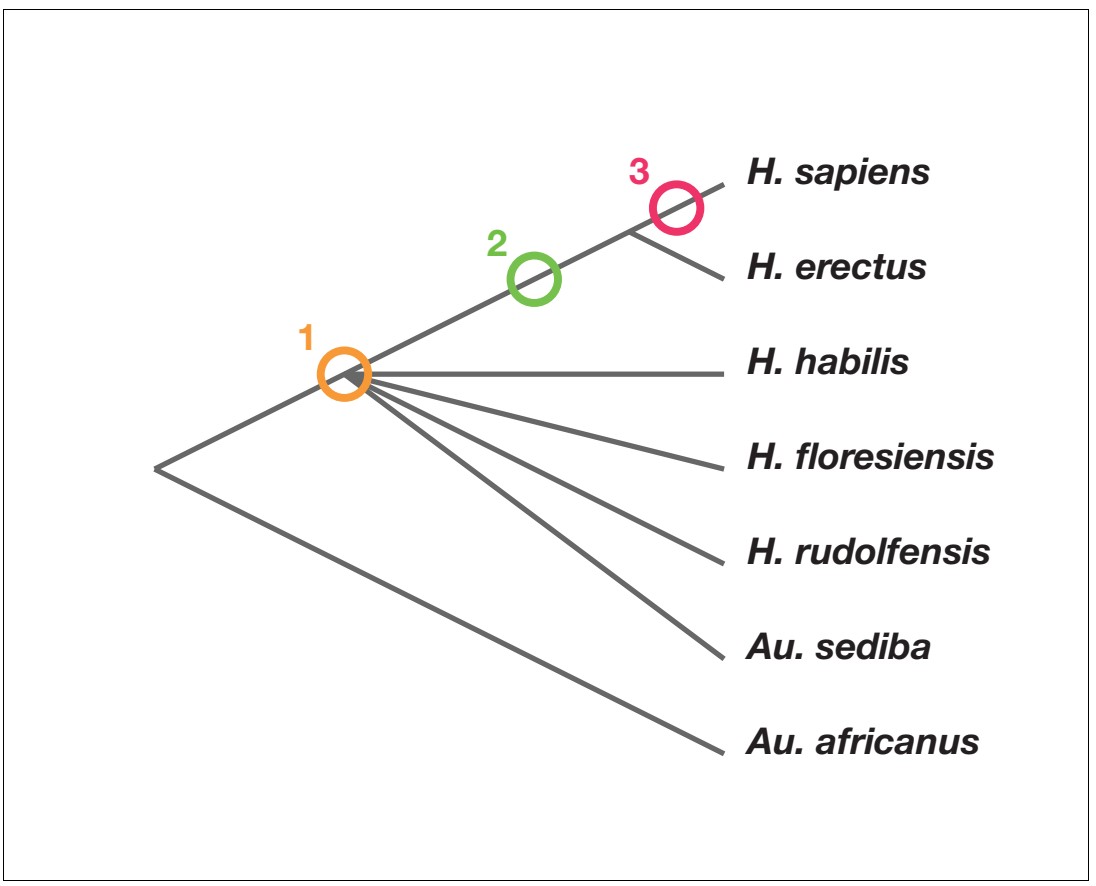

**Figure 2.** Phylogenetic scenarios for *H. naledi*. A simplified cladogram of *Homo*, with the possible placements of *H. naledi* indicated. The cladogram places *A. africanus* as an outgroup to the *Homo + Au. sediba* clade, as consistent with nearly all phylogenetic analyses of these species (*Berger et al., 2010*; *Dembo et al., 2015*, *2016*). To simplify the tree, we have omitted *H. antecessor*, *H. heidelbergensis* and Neanderthals, which all phylogenetic analyses place as sisters to *H. sapiens* relative to *H. erectus*. There is no present consensus about the branching order among *H. habilis*, *H. rudolfensis*, *H. floresiensis* and *Au. sediba* (*Dembo et al., 2015*, *2016*), and so these are depicted as a polytomy.

pattern in this species (*Laird et al., 2017*), although it seems likely that postcranial features would be subject to greater adaptive constraints.

An alternative hypothesis for the homoplastic appearance of *H. naledi* is hybridization among two or more hominin lineages. As ancient DNA evidence has grown, it has become clear that hybridization among genetically distant human lineages occurred many times (*Kuhlwilm et al., 2016*; *Meyer et al., 2012*; *Prüfer et al., 2014*), as is the case in chimpanzees and bonobos (*de Manuel et al., 2016*) and in many other mammalian lineages (*Schaefer et al., 2016*). The mosaic anatomy of *H. naledi*, which includes many shared derived characters of modern humans and *H. erectus*, might suggest the hypothesis that *H. naledi* resulted from the hybridization of a more human-like population and a late-surviving australopith. This hypothesis remains untestable with the current evidence, although it seems more parsimonious to suggest that *H. naledi* itself survived from an early period of diversification of *Homo*. Morphology does not rule out the possibility that *H. naledi* originated in the Early Pleistocene as a result of the hybridization of different populations, and persisted long after this hybrid speciation. The evidence of genetic mixture among more recent hominins makes this hypothesis seem reasonable, but again it is untestable unless genetic material is obtained from the fossils. Attempts to obtain aDNA from *H. naledi* remains have thus far proven unsuccessful.

In addition, we have reported several apparent autapomorphies that are present across the skeleton of *H. naledi*. These include the morphology of the thumb, aspects of the morphology of the

spine, and aspects of the morphology of the proximal femur (*Berger et al., 2015*; *Kivell et al., 2015*; *Marchi et al., 2017*). Unfortunately, the Pliocene hominin record is poor, and without clearly understanding the ancestral lineage of *H. naledi*, and whether we have in fact already discovered its ancestors, we cannot know whether such features may have been present in the last common ancestor (LCA) of *H. naledi* and other hominin species, and are thus actually primitive in *H. naledi*'s lineage rather than uniquely derived. Therefore, the importance of these apparent autapomorphies in establishing the origins of *H. naledi* remain unresolved.

## Implications for the fossil record

Until now, palaeoanthropologists and archaeologists have generally assumed that morphologically primitive hominins such as *H. naledi* did not survive into the later parts of the Pleistocene in Africa. This assumption has guided the interpretation of fossil discoveries with poor geological or stratigraphic context, including the many surface finds that make up the majority of the record from ancient lacustrine and riverine deposits (e.g. *Taieb et al., 1976*; *Yuretich, 1979*; *Kalb et al., 1982*; *Tiercelin, 1986*; *Ward et al., 1999*; *WoldeGabriel et al., 2001*; *Clark et al., 2003*; *Gathogo and Brown, 2006*; *McDougall and Brown, 2006*; *Campisano and Feibel, 2008*; *Campisano, 2012*). These and other studies have shown that in many African sedimentary contexts, Pliocene or Early Pleistocene sediments are overlain by deposits of Middle or Late Pleistocene age or even by Holocene-aged deposits. It is common knowledge that fragmentary fossils of Plio-Pleistocene age occur ex situ on the surface with Middle Stone Age (MSA), Later Stone Age (LSA), or historic artifacts; in the absence of in situ association, anthropologists often rely upon a fossil's morphology as an indicator of its age.

The discovery of *H. naledi* provided a natural experiment to test whether anthropologists can reliably establish the approximate age of hominin fossil fragments from their morphology. Before the publication of a geological age for *H. naledi*, many anthropologists examined its entire morphological pattern and concluded that the species must date to more than 1.5 million years ago. This includes one formal morphological study (*Thackeray, 2015*) and many other published comments by experts. A second study concluded that the Dinaledi hominin sample could be 930,000 years old, though the confidence interval on this estimate ranged from the present to c. 2.5 Ma (*Dembo et al., 2016*).

These examples show that expert intuition about the ages of fossil samples is likely to be wrong when based on their morphology alone. We must therefore demand fuller information about the geological context both of surface finds and of finds that are reported as in situ. If fragments of *H. naledi* had been found in isolation—instead of in the cohesive assemblage of the Dinaledi Chamber—many parts of its anatomy individually may have been confused for hominin material of Pliocene age. As we have noted, parts of the *H. naledi* cranial vault, dentition, shoulder, manual phalanges, pelvis and proximal femur would be easily misattributed to *Australopithecus*. Other parts of the hand, dentition, foot, and lower limb exhibit morphology similar to that of modern humans or *H. erectus*. As we know neither the origination point nor the extinction time of *H. naledi*, it is conceivable that fragments from this species have already been misattributed to other hominin taxa.

## Implications for the archaeological record

*H. naledi* has traits that were long considered to be adaptations for creating material culture. Its wrist, hand and fingertip morphology share several derived features with Neanderthals and modern humans that are absent in *H. habilis*, *H. floresiensis*, and *Au. sediba* (*Kivell et al., 2015*). If these features evolved to support habitual tool manufacture in Neanderthals and modern humans, then it is reasonable to conclude that *H. naledi* was also fully competent in using tools. The use of tools and the consumption of higher-quality foodstuffs including meat and processed plant resources have been hypothesized as evolutionary pressures leading to dental reduction in hominins (*Zink and Lieberman, 2016*). The small dentition of *H. naledi* manifests this adaptive strategy to a greater extent than *H. habilis*, *H. rudolfensis* and most *H. erectus* samples (*Berger et al., 2015*; *Hawks et al., 2017*), though without the predicted encephalization.

What tools did *H. naledi* make? Its lineage may have existed across much or all of the time during which African hominin populations were manufacturing Acheulean and possibly even Oldowan assemblages (e.g. *Mcbrearty and Brooks, 2000*). The *H. naledi* lineage also existed during at

least the first half of the MSA, which as an archaeological category seems to have commenced more than 400 ka in several instances in subequatorial and northeastern Africa (*Dusseldorp et al., 2013*; *Wilkins and Chazan, 2012*; *McBrearty and Tryon, 2006*; *Mcbrearty and Brooks, 2000*).

Many previous workers have grappled with the question of which hominin species were the makers of Early Stone Age industries (e.g. *Foley, 1987*; *Susman, 1991*; *Domalain et al., 2016*). A key part of these considerations has been the role of brain size and behavioural ecology in sustaining traditions, which have supported the role of larger-brained *H. habilis* and *H. erectus* as toolmakers and have downplayed the possibility that small-brained *Paranthropus* may likewise have innovated (e.g. *Hopkinson et al., 2013*; *Domalain et al., 2016*). With some exceptions (e.g. *Stringer, 2011*), there has been a widespread assumption that MSA traditions were made by modern humans or their ancestors, whether denoted as 'archaic *H. sapiens*' or as a precursor such as '*H. helmei*' (*Mcbrearty and Brooks, 2000*; *Lahr and Foley, 2001*; *Stringer, 2002*; *Henshilwood and Marean, 2003*; *Henshilwood and Marean, 2006*; *Dusseldorp et al., 2013*). MSA variants are characterized by the manufacture of blades, by the presence of the Levallois flaking technique and of hafted implements, at some locations by the use of pigments, and by a lack of emphasis on large cutting tools such as the handaxes and cleavers of the Acheulean industry (e.g. *Mcbrearty and Brooks, 2000*; *Henshilwood and Marean, 2003*; *Marean and Assefa, 2005*; *Henshilwood and Marean, 2006*). Some of these technical innovations have even been considered as markers of modern human behaviour.

However, it is now clear that the populations of subequatorial Africa had deep prehistoric divisions (*Stringer, 2016*; *Lachance et al., 2012*; *Hsieh et al., 2016*) and that multiple genetically and morphologically divergent hominin populations probably created Acheulean and MSA archaeological traditions. This situation is paralleled outside of Africa, where most of the manufacturing techniques that characterize the MSA were also mastered by Neanderthals and possibly by Denisovans (*Roebroeks and Soressi, 2016*; *d'Errico and Banks, 2013* ). These archaic populations diverged from African populations well before the appearance of such techniques either in Africa or in Eurasia (*Meyer et al., 2016*), so these techniques must either have been invented independently multiple times or have been transferred by long-distance exchange of ideas across long-separated hominin populations.

*H. naledi* existed contemporaneously with MSA archaeological industries across subequatorial and northeastern Africa (*Mcbrearty and Brooks, 2000*; *Henshilwood and Marean, 2003*, *Marean, 2006*; *Marean and Assefa, 2005*; *Henshilwood and Marean, 2006*; *McBrearty and Tryon, 2006*; *Wilkins and Chazan, 2012*; *Dusseldorp et al., 2013*; *Wurz, 2013*). Excavations in the Rising Star cave system have not yet uncovered artifacts in direct association with *H. naledi*. But considering the weak nature of the fossil hominin record, *H. naledi* may be the only hominin definitely known to be present during at least the early part of the MSA in the highvelt region of southern Africa (*Dusseldorp et al., 2013*). Considering the context, it is possible that *H. naledi* sustained MSA traditions. Without extraordinary evidence, we cannot uncritically accept that such a broadly defined archaeological tradition was the exclusive product of a single population across Africa.

## Possible evidence for mortuary behaviour

Did *H. naledi* deliberately deposit bodies within the Rising Star cave system? With respect to the deposition of the fossil material, it is appropriate to adopt a null hypothesis that the remains entered the Dinaledi and Lesedi Chambers without intentional hominin mediation, and to see whether the evidence can reject that hypothesis. We have previously examined depositional scenarios on the basis of evidence from the Dinaledi Chamber (*Dirks et al., 2015*, *2016*; *Randolph-Quinney et al., 2016*). The discovery of hominin material in the Lesedi Chamber adds a second instance of deposition of hominin skeletal material within the cave system.

Some other cave systems in the Cradle of Humankind area likewise present evidence of multiple episodes of the deposition of hominin remains. Swartkrans has a complex series of infills that contain hominin and a broad array of macrofaunal remains, many of which bear evidence of carnivore or scavenger activity representing multiple accumulating agents (*Pickering et al., 2004a*). Further, evidence of cutmarks, percussion marks, and burned bone show that hominins were an accumulating agent of some Swartkrans faunal remains (*Pickering et al., 2005*). Sterkfontein is another cave system that has a complex series of infills, in which much bone material bears traces of carnivore and

scavenger activity. Within Sterkfontein, the Silberberg Grotto is a deep chamber that contains one hominin skeleton (StW 573) together with faunal remains that appear to have fallen from above; it is a death trap (*Pickering et al., 2004a*). Also within the Sterkfontein system, the Jacovec Cavern breccia presents some evidence for water transport of material from the surface and water sorting of bone (*Kibii, 2007*). These examples provide several hypotheses for the deposition of hominin skeletal remains that do not involve intentional behaviour by the hominins themselves, and we have previously examined whether the Dinaledi Chamber evidence is compatible with any of them (*Dirks et al., 2015*, *2016*).

While geological and sedimentological studies of the Lesedi Chamber are still ongoing, we can consider how its taphonomic situation resembles the Dinaledi Chamber material. In the Dinaledi Chamber, the skeletal material showed invertebrate surface modification but a complete lack of markings from carnivores, scavengers, or hominins (*Dirks et al., 2015*, *2016*). The Lesedi Chamber hominin material likewise presents no evidence of cutmarks, tooth marks, scoring, puncture marks, gnawing or bone cylinders, and only shows surface markings consistent with abrasion or pitting, many after the deposition of manganese and iron oxide coatings on the bones (*Hawks et al., 2017*). These observations seem to exclude carnivores and scavengers as the primary accumulating agents for the assemblages.

The Dinaledi Chamber is enormously challenging to reach today, and both sedimentological and geological evidence supports the hypothesis that the chamber itself and the entry chute from the neighboring Dragon's Back Chamber had substantially the same configuration at the time at which the *H. naledi* skeletal remains entered (*Dirks et al., 2015*, *2016*). Some have questioned whether one or more alternative entrances to the Dinaledi Chamber may once have existed, which might have made the physical situation much easier for *H. naledi* to enter the chamber from the outside (*Val, 2016*; *Thackeray, 2015*). But any such entrance would have needed to replicate most of the constraints of the present entrance, or else it would not produce the sedimentological distinctiveness of the Dinaledi Chamber or the lack of non-hominin macrofauna (*Dirks et al., 2016*; *Randolph-Quinney et al., 2016*). The situation in the Lesedi Chamber makes these constraints of the Dinaledi Chamber even more apparent. The Lesedi Chamber is similarly situated deep inside the cave system, far inside the dark zone, with no nearby surface entrance (*Hawks et al., 2017*). However, no strong physical constraint prevents macrofauna, at least those smaller than humans, from entering. Faunal material in the chamber demonstrates that at least the remains of small carnivores and smaller fauna did enter the Lesedi Chamber, even though it is deep in the cave, well within the dark zone. Although we do not know the timing or manner in which these faunal elements entered the Lesedi Chamber, their presence reinforces the importance of physical constraints in impeding entry into the Dinaledi Chamber, where no such faunal remains have been found (*Dirks et al., 2015*). Further sedimentological and geological assessment of the Lesedi Chamber, and direct dating of the faunal and hominin remains, may clarify the relation of faunal and hominin remains.

*Val (2016)* proposed that the hominin skeletal material from the Dinaledi Chamber may have been transported from another location within the cave system, which we have not located, but which might itself have been consistent with carnivore accumulation or a death trap from the surface. In Sterkfontein, there may have been redeposition of sediments from higher chambers into the Silberberg Grotto (*Kramers and Dirks, 2017*), providing a possible example a process driven by gravity from above, although the StW 573 skeleton itself appears to be in near-primary context. No openings in the ceilings above the Dinaledi or Lesedi Chambers appear consistent with the gravity-driven transport of material from directly above. The Dinaledi Chamber skeletal material shows no evidence of high-energy fluvial transport, which would have been necessary to move such a quantity of bone any horizontal distance through the cave (*Dirks et al., 2015*, *2016*). The same is true of the remains within the Lesedi Chamber (*Hawks et al., 2017*). In both deposits, there is evidence of post-depositional reworking of sediments, but in both deposits, some articulated remains have been recovered, and neither the skeletal element representation nor the physical condition of the remains are consistent with wholesale secondary redeposition of the hominin assemblages from any third location (*Dirks et al., 2016*; *Hawks et al., 2017*).

We consider it untenable to hypothesize that both the Dinaledi Chamber and the Lesedi Chamber were accidental death trap situations. We have previously written (*Dirks et al., 2015*, *2016*) that the accidental death trap hypothesis was one that the physical evidence from the Dinaledi Chamber

might not reject. Still, the evidence that hominin individuals of all ages were deposited in the chamber over some period of time, as well as the sediment composition within the chamber itself, led us to view that hypothesis as less likely. Other cave systems in the region have been hypothesized as death-trap situations, including Malapa (*Dirks et al., 2010*; *L'Abbé et al., 2015*; *Val et al., 2015*) and the Silberg Grotto of Sterkfontein (*Pickering et al., 2004a*). In these instances, a relatively direct vertical route existed either from the surface or from cave chambers above; furthermore, other non-cryptic macrofauna were present among the fossil remains, externally derived sediments were abundant, and (in the case of Malapa) plant remains were also present. No such evidence occurs in the Dinaledi Chamber. As we continue to study the geological history of the Rising Star system, it is possible that we will find that *H. naledi* or small carnivores accessed the Lesedi Chamber differently than excavators today, but we have found no evidence of a nearby vertical entry that would accommodate hominins. The presence of two such situations in the same cave system, with no remaining evidence of a death trap other than the hominin remains, would be unlikely.

The evidence in its entirety appears incompatible with the hypothesis of accumulation in both the DInaledi and Lesedi Chambers without some hominin agency. However, with a later Middle Pleistocene date for the Dinaledi Chamber material, we must also consider the suggestion that modern humans or their immediate predecessors were accumulating agents for the *H. naledi* skeletal material (e.g., C. Marean, quoted in *Gibbons (2015)*. This hypothesis would require that the modern humans left no cutmarks or tooth marks on the *H. naledi* material, and that they treated *H. naledi* remains differently than those of any other species, including modern humans themselves. Also, although it is possible that *H. naledi* may have existed in contact with ancestors of modern humans, we have as yet no evidence of this. We will continue to explore the possible interactions of *H. naledi* and other hominin populations, but they do not appear to be a likely explanation for the deposition of skeletal remains in the Rising Star cave system.

We propose that funerary caching by *H. naledi* is a reasonable explanation for the presence of remains in the Dinaledi and Lesedi Chambers. Mortuary behaviours, while culturally diverse, are universal among modern human cultural groups (*Pettitt, 2010*). Such behaviours are not seen in living non-human primates or in other social mammals, but many social mammals exhibit signs of grief, distress, or other emotional response when other individuals within their social group die (*King, 2013*). We have no information about whether *H. naledi* was a symbolic species, although with the possibility that it manufactured MSA toolkits, we do not rule out such abilities. But symbolic cognition is not likely to have been necessary to sustain a repeated cultural practice in response to the physical and social effects of the deaths of group members (*Pettitt, 2010*). Such behaviour may have many different motivations, from the removal of decaying bodies from habitation areas, to the prevention of scavenger activity, to social bonding, which are not mutually exclusive. We suggest only that such cultural behaviour may have been within the capabilities of a species that otherwise presents every appearance of technical and subsistence strategies that were common across the genus *Homo*.

## Conclusions

Fossil and genetic evidence shows that subequatorial Africa was home to diverse populations of hominins throughout the Pleistocene. The expansive savanna and open woodland habitats of this region have driven biodiversity in many mammals and birds that have similar habitat preferences to hominins (e.g. *Bannerman and Burns, 1953*; *Kingdon, 2015*; *Payne, 2013*). We suggest that for hominin populations too, subequatorial Africa appears to have been a source of biological diversity and innovation. No paleoanthropologist anticipated that a species like *H. naledi* existed in this region during the late Middle Pleistocene. However, considering a broad array of biogeographic, phylogenetic, and genetic evidence from humans and other mammals, the discovery of more members of a diverse community of hominin populations in this vast region should no longer be a surprise.

This hypothesis should provoke greater examination of the paleoenvironments and regional paleoclimate across this region of Africa. Further, the presence of a diversity of hominin populations throughout most of the Pleistocene must lend caution to how we examine fragmentary specimens. Many of these populations and species are indistinguishable from each other in many parts of the skeleton, despite being very different in others. In particular, we must apply renewed caution to

behavioural inferences. The transition to early MSA industries is likely to have involved a broad array of hominin populations and/or species that possibly interacted with each other.

Is *H. naledi* the direct ancestor of humans or of any other hominins? The later Middle Pleistocene age of the Dinaledi Chamber assemblage is substantially later than the date presently recognized for the first appearance of *H. erectus* some 1.8 million years ago. The Dinaledi occurrence of *H. naledi* is later in time than the hypothesised genetic divergence of Neanderthal and modern human populations more than 500,000 years ago (*Meyer et al., 2016*). But a paleontological view recognizes that any particular set of fossils does not represent the entire time depth of a species or its relationships, and phylogenetic analyses of *H. naledi* show that the species and its branch must have existed much earlier than the Dinaledi fossils (*Dembo et al., 2016*; *Hawks and Berger, 2016*; *Thackeray, 2015*). One analysis of craniodental evidence places *H. naledi* amid the branches leading to *H. habilis*, *Au. sediba*, and *H. rudolfensis*, suggesting that its anatomical pattern may have been present from the earliest origin of *Homo*. Another analysis has placed *H. naledi* as a sister taxon to archaic species of *Homo* and modern humans, closer to living humans than *H. erectus* (*Dembo et al., 2016*). If this is true, an early *H. naledi* population may have been the ancestor of humans, placing *H. erectus* as a side branch.

*H. naledi* is clearly a primitive species within the genus *Homo*, despite sharing many derived features with archaic and modern humans. The fossil record for other species attributed to early *Homo* is presently too incomplete to ascertain whether these species also show such mosaicism, or whether they express different manifestations of primitive and derived morphological patterns. A species like *H. naledi* might well have given rise repeatedly to other branches of *Homo*, each derived in a somewhat different way. A fresh look at the hominin fossil record, setting aside a history of linear assumptions about the evolution of *H. erectus* and *H. sapiens*, may set a new context for further fossil discoveries. Better analytical techniques, and increased knowledge provided by aDNA, may shed further light on these questions.

## Acknowledgements

We would like to thank the many funding agencies that supported various aspects of this work. In particular, we would like to thank the National Geographic Society, the National Research Foundation of South Africa and the Lyda Hill Foundation for significant funding of the discovery, recovery and initial analysis of this material. Further support was provided by the Australian Research Council ARC (DP140104282: PHGMD, ER, HHW). We would also like to thank the University of the Witwatersrand and the Evolutionary Studies Institute, as well as the South African National Centre of Excellence in PalaeoSciences. We would like to thank the South African Heritage Resource Agency for the necessary permits to work on the Rising Star site, and the Jacobs family and later the Lee R Berger Foundation for granting access. The assistance of members of the Speleological Exploration Club, in various mapping exercises and with safety during excavations, is gratefully acknowledged. We would also like to thank Wilma Lawrence, Bonita De Klerk, Natasha Barbolini, Merrill van der Walt, Wayne Crichton, Justin Mukanku, Sonia Sequiera and Wilhelmina Pretorius for their assistance during all phases of the project.

## Additional information

### Funding

| Funder | Grant reference number | Author |
|---|---|---|
| National Geographic Society | | Lee R Berger |
| National Research Foundation | | Lee R Berger |
| Lyda Hill Foundation | | Lee R Berger |
| Australian Research Council | DP140104282 | Paul HGM Dirks<br>Eric M Roberts |

The funders had no role in study design, data collection and interpretation, or the decision to submit the work for publication.

## Author contributions

LRB, Conceptualization, Resources, Data curation, Formal analysis, Supervision, Funding acquisition, Investigation, Writing—original draft, Project administration, Writing—review and editing; JH, Conceptualization, Data curation, Formal analysis, Validation, Investigation, Visualization, Writing—original draft, Project administration, Writing—review and editing; PHGMD, ME, Validation, Investigation, Writing—review and editing; EMR, Funding acquisition, Validation, Investigation, Writing—review and editing

## Author ORCIDs

Lee R Berger, http://orcid.org/0000-0002-0367-7629
Paul HGM Dirks, http://orcid.org/0000-0002-1582-1405

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
