## [Decision Letter]

Thank you for submitting your article "*Homo naledi* and Pleistocene hominin evolution in subequatorial Africa" for consideration by eLife. Your article has been reviewed by three peer reviewers, and the evaluation has been overseen by a Reviewing Editor and Ian Baldwin as the Senior Editor. One of the three reviewers, Chris Stringer, has agreed to share his name. The other reviewers remain anonymous.

The reviewers have discussed the reviews with one another and the Reviewing Editor has drafted this decision to help you prepare a revised submission.

This paper aims to summarize (and extend beyond) the current state of knowledge on *Homo naledi* based on the initial publications that described the initial material, the subsequent analyses of that material that have been published since, and now including the new discoveries and analyses presented in the manuscripts currently under consideration at *eLife*. The concept of the present, synthesis paper has a place as a part of the set of current articles, and could be developed into an excellent contribution that will be valuable to readers.

However, the reviewers expressed the consensus view (confirmed during the consultation process) that this paper should focus more heavily on summarizing the overall results and main messages of the research performed, and de-emphasize the presentation of scenarios that are not currently sufficiently supported by evidence. In particular, the reviewers pointed to the idea of *H. naledi* as a wide-ranging species, to that of the controlled use of fire by *H. naledi*, and to the deliberate disposal/burial scenario, as among those that cross this line. Details of these critiques, and other issues that need to be addressed before the manuscript can be considered further, are provided as part of the full reviews, below. Note that one reviewer also provided detailed comments in the attached document.

There is definitely room for some speculation following a thorough summary (especially in the context of the surprisingly recent dates), but this speculation should be explicitly marked, expressed with more caution and with an unbiased treatment of the evidence at hand, and perhaps more limited to a subset of the scenarios currently discussed.

We will likely ask reviewers to evaluate the revised manuscript before considering it for acceptance, and this process could be iterative given the dependency of the present manuscript on others.

*Reviewer #1:*

This is the synthesis paper for the suite of *H. naledi* papers that suggests that *H. naledi* was not a limited, geographically restricted population, that it may have been at the base of the *Homo* lineage and led to *H. sapiens*, and that it practiced deliberate burial of its members. I think that perhaps the authors may have interpreted beyond the data available.

First, their claim that the anatomy of *H. naledi* suggests it was a wide-ranging species. They are correct that there is no evidence to refute it, but there is also no evidence to support it. The Rising Star cave system is the only place where specimens of *H. naledi* have been recovered. They certainly did not live in the cave and it would be reasonable to assume that they came from the area surrounding the cave system, but beyond that, we have no evidence that they were anywhere else.

Second, the authors suggest that based on the morphology of *H. naledi*, it is a primitive member of the *Homo* lineage with some derived features shared with "archaic" and modern humans. This is used to suggest that *H. naledi* led to modern humans without elaboration on the traits. For example, why are some traits weighed more than others? Which traits unite *H. naledi* with modern humans exclusively? Why are they not convergences, etc? The authors also suggest that the geological age of *H. naledi* does not matter in terms of interpreting its phylogenetic relationship. I think that many, if not most, paleontologists would disagree with that statement. Geological age does not need to be (nor should it be) a consideration when considering taxonomy, but it is crucial for interpreting the meaning of the morphology and certainly for reconstructing phylogenetic relationships, particularly when we are talking about the complex realities of Pleistocene hominin evolution.

Third, every paper in this suite of submission tries to advance the hypothesis of deliberate burial behavior in *H. naledi*. I actually do not have strong feelings regarding the existence of mortuary practices in *H. naledi*. However, I do not think that the authors have provided sufficient evidence to reach their conclusion. The papers read as a concerted effort to find data and interpretations of the data that fit the pre-existing hypothesis of deliberate burial rather than independently assessing the data to see if they support or reject the hypothesis.

*Reviewer #2:*

This is a highly speculative paper (e.g., hybridization hypotheses), although quite entertaining. I suppose it functions well to introduce the rest of the new papers of Berger's team presenting new *H. naledi* fossils from the Rising Star cave system.

These are some of my minor comments to consider:

- Please revise inconsistent use of capitalizations and commas along the text.

- Lines 40, 182: The authors make the case that *Homo naledi* could be from early *Homo* stock in the Pliocene, based on the analyses by Dembo et al. *But* those analyses are heavily influenced by the small brain size of the original *naledi* finds. How do the authors expect this will change with the larger brain size estimates? What does this mean for the phylogenetic hypotheses they offer?

- Starting in line 47: please provide references for each of these 'African Middle Pleistocene' 'extreme sparse' hominin materials. There is not a single reference in this paragraph.

- Lines 63-64: Could the authors please provide some discussion about morphological similarities between the Florisbad molar (likely associated with the skull) and that of *Homo naledi*? Not just a size comparison. Is it possible these are same species, or is the morphology different?

- Lines 139-140: "We now include a small-brained species of hominin in this recognized diversity." Are we talking about *Homo floresiensis* or *naledi*? Why isn't *H. floresiensis* discussed here?

- Starting in lines 164: I do not know any study inferring endemism/geographical range based on morphology. Please consider rewording this section.

- Lines 184-187: 'An informal consideration of postcranial traits suggests that they might support scenario 2, as *H. naledi* shares many derived features of the hand, foot, and lower limb with *H. erectus* and *H. sapiens* that are absent from *H. habilis, H. floresiensis*, or *A. sediba*, yet lacks several derived traits of the shoulder, trunk, and hip shared by *H. erectus* and *H. sapiens*.'

Please be fair to the current state of the knowledge: We don't really know what the hand, foot or lower limb of *H. habilis* looked like. Both the OH7 hand and the OH8 foot could belong to another taxon (e.g., Wood 1974 JHE, Wood and Constantino Yearbook Phys Anth, Moyà-Solà et al 2008 Folia Primatol).

And in terms of it lacking "several derived traits of the shoulder, trunk, and hip shared by *H. erectus* and *H. sapiens*", please be specific here.

*Reviewer #3:*

In my view this paper should be suitable for publication after minor revisions.

The references for all four papers need work for consistency of formatting, journal abbreviations, accents etc.

"An obvious question is how the *H. naledi* lineage fit into the hominin diversity of the Early and Middle Pleistocene..." If terms like early and middle are being used informally, they should not be capitalised?

"This single tooth is assumed to be associated with the [Florisbad] fossil hominin cranium." Well this does seem a reasonable assumption?

"Some workers have argued that the Kabwe skull may be 300 ka or earlier (Bräuer 2008)." See Stringer AAPA abstract 2011 and Lone Survivors/The Origin of our Species for discussion of ongoing work on the dating of the cranium.

"…both of Middle Pleistocene age". How well-dated is Berg Aukas?

"Today's populations are descendants of relatively large and stable human populations across most of the last 200,000 years..." In what sense stable?

"The full geographic extent of *H. naledi* is unknown, but its anatomy shows it was unlikely to have been an endemic species limited to any small part of subequatorial Africa." I'm not sure anatomical arguments are sufficient to make this point...

"The evidence of genetic mixture among more recent hominins makes this hypothesis seem reasonable but likewise untestable." In my commentary last year I argued that aDNA work on the Sima fossils held promise for the *naledi* material – can the authors comment on any ongoing work?

"A second study concluded the Dinaledi hominin sample [*could be about*] 930,000 years old..."

"Its definition is based on the manufacture of blades, presence of the levallois flaking technique, hafted implements, use of pigments..." There are plenty of MSA sites that don't have evidence of pigments

"Until now, archaeologists investigating the Middle Stone Age have largely assumed that all African fossil hominins from the period were part of a single variable species that was in a broad sense ancestral to modern humans." Also see my 2011 AAPA abstract: The chronological and evolutionary position of the Broken Hill cranium.

"This situation is paralleled outside of Africa, where most of the manufacturing techniques of the MSA were also mastered by Neanderthals and Denisovans'." Given the Neanderthal presence at the site as well, how can Denisovan archaeology be recognised?

"Considering the context, it is [*possible*] that *H. naledi* sustained MSA traditions."

"substantially later than the presently recognized first appearance date of H. erectus some 1.8 [GIVEN DMANISI] million years ago"

"(Dembo et al. 2016). If this is true, an early H. naledi population may have been the ancestor of [*modern*] humans..." Can't see the Dembo et al. refs...and insert modern here.

*Reviewer #3 Minor Comments:*

I have added suggested edits to a text version of the paper (attached).

[Further changes were requested before acceptance.]

Thank you for resubmitting your work entitled "*Homo naledi* and Pleistocene hominin evolution in subequatorial Africa" for further consideration at eLife. Your revised article has been evaluated a Reviewing Editor and three reviewers.

The manuscript has been improved but there are some remaining issues that need to be addressed before acceptance, as outlined below:

The major, outstanding consensus issue from the reviewers concerns discussion within the paper on the mortuary practices of *Homo naledi*. This comment is of course related to the decision letter on the linked article "New fossil remains of *Homo naledi* from the Lesedi Chamber, South Africa" and the limited confidence reflected therein in the evidence thought to support the deliberate burial conclusion.

From an editorial perspective, the format and synthesis nature of the present article does provide some leeway for discussing diverse hypotheses and areas of investigation and interest to the authors, including related to ongoing questions concerning the *H. naledi* deliberate burial scenario. That said, and taking into account the reviewer consensus on the evidence presented in the linked paper, the discussion of non-deliberate burial scenarios deserve more thorough treatment (e.g., beyond the unlikely prolonged mass casualty event) and weight than they are given at present. Ideally, non-deliberate burial would be treated as the null hypothesis, with careful consideration of whether that hypothesis could be rejected. Providing a clearly marked opinion that the authors favor the deliberate burial scenario is then fine. The editor and reviewers agree with the authors that it should not be presumed that only large-brained hominins have the capacity for deliberate burial; however, the consensus view is that this behavior has not yet been sufficiently demonstrated for *H. naledi*.

The reviewers also had some discussion with respect to the consideration of the evolutionary scenarios in the "What explains the mosaic anatomy of *H. naledi*" section. Absent in the present version of the manuscript is reference to and relevant discussion of a previous *H. naledi* paper by Laird et al. (2017 Journal of Human Evolution, on which authors from the present submission are co-authors) that examines evolutionary scenarios under various chronological frameworks and offers differing interpretations with respect to gene flow. This should be rectified and the scope of the discussion in this section expanded accordingly.

The reviewers also identified multiple more detailed issues with the present submission that need to be addressed. These are provided below. One reviewer provided additional comments (including notation of a subset of the citations that appear to be missing from the reference list) in a file that is attached to this decision letter.

Detailed Issues:

1) "Among the better preserved are three partial crania from Morocco, five from Ethiopia, one from Kenya, two from Tanzania, one from Zambia and possibly two from South Africa." I'm not always sure which are being referred to, but there are at least 2 from Kenya?

2) "Genetic evidence shows that equally diverse populations of archaic humans once existed in subequatorial Africa (Hammer et al. 2011; Stringer, 2011; Lachance et al. 2012), and although no fossil evidence can yet be attributed to these groups, the Middle Pleistocene record of this region does speak to the presence of diversity." Only sparse evidence? You've mentioned Broken Hill already and, for example, Eliye Springs looks very different to Omo 2.

3) "*H. naledi* existed contemporaneously with Middle Stone Age archaeological industries across subequatorial and north eastern Africa." True even if we took the highest age range figure i.e. 420 ka?

4) "Further discoveries of *H. naledi* may show it also survived into the periods of later MSA [or even LSA] traditions." I think the LSA bit is a speculation too far, at present!

5) "The later Middle Pleistocene age of the Dinaledi Chamber assemblage is substantially later than the presently recognized first appearance date of H. erectus some 1.75 million years ago." As I commented previously, doesn't Dmanisi age probably exceed this figure?

---

## [Author Response]

We found the reviews and comments helpful and constructive and have taken all of the reviewers and editorial concerns on board.

*Minor comments*

We have dealt with the minor issues of inconsistency in capitalization, commas, formatting, references etc. throughout the manuscript. We have corrected each minor point indicated by Reviewers #2 and #3 (Reviewer #1’s comments were more general and are dealt with below). We have included more detailed references to other hominin material discussed in the manuscript.

We have attempted to be more specific, through reference citation and comment throughout concerning the state of knowledge of the postcranial anatomy of other species of early *Homo* as well as where we reference primitive and derived traits.

We have altered sentence structure and wording as suggested by Referees #2 and #3 where possible, and have altered sentences for clarification where the meaning was unclear.

*Major comments*

“The reviewers expressed the consensus view (confirmed during the consultation process) that this paper should focus more heavily on summarizing the overall results and main messages of the research performed, and de-emphasize the presentation of scenarios that are not currently sufficiently supported by evidence. In particular, the reviewers pointed to the idea of *H. naledi* as a wide-ranging species, to that of the controlled use of fire by *H. naledi*, and to the deliberate disposal/burial scenario”.

Reviewer #2 also questions the morphological endemism/geographical range argument. We have attempted to concentrate on matters objectively established in the other two manuscripts (parts of the Elliott et al manuscript having now been combined with the Hawks et al manuscript). We have removed the speculation, based on lower limb anatomy, that *H. naledi* was a wide-ranging species. We have used caution in our wording concerning the potential for geographic distribution of the species. We have removed speculation about the use of fire.

We have used greater caution in our discussion of the evidence for mortuary behaviour by *H. naledi*. We have not deleted this discussion as a second chamber does in some ways add evidence to the presence of an unusual accumulating situation beyond that typical for other hominin sites (and we believe this to be of great interest to the readers), but we have taken special care to go no further than to express our opinion that “we feel it is unlikely to appear in multiple geologically independent occurrences.” This, and other more cautious statements throughout the manuscript and in this section, should allow the reader to weigh the weight of evidence one way or the other on the matter as presented now in Hawks et al. and with minimal speculation on our part. Given that the Elliott manuscript is now withdrawn, we would hope that the suite of papers no longer reads like “as a concerted effort to find data and interpretations of the data that fit the pre-existing hypothesis of deliberate burial”. That was not our intent.

Concerning the comments of Reviewer #2 on the hybridization hypotheses as “speculative”, we feel that given the strong genetic evidence from aDNA emerging that hybridization is not uncommon in the hominin record, and that indeed the Hawks et al paper supports further that *H. naledi* indeed shows a mosaic (surprisingly so) suite of primitive and (very) derived characters, this discussion is valuable to the reader. We have though attempted to soften the speculative nature of our discussion, while keeping this section in the manuscript.

We have attempted to reduce all other use of speculative terminology throughout the manuscript. We have also (as suggested by Reviewer #1) more thoroughly referenced the presence of primitive and derived characters in the skeleton of *H. naledi*. We have also elaborated on these traits both in this manuscript and the Hawks et al manuscript through specific mention and through citation. We have also dealt more thoroughly (and cautiously) with the origins of the mosaic anatomy of *H. naledi* throughout the discussion, elaborating on what we know, and what we simply do not know, especially with regard to the potential for convergences, homoplasy’s etc.

Concerning Reviewer #1’s comment *“The authors also suggest that the geological age of H. naledi does not matter in terms of interpreting its phylogenetic relationship. I think that many, if not most, paleontologists would disagree with that statement. Geological age does not need to be (nor should it be) a consideration when considering taxonomy, but it is crucial for interpreting the meaning of the morphology and certainly for reconstructing phylogenetic relationships, particularly when we are talking about the complex realities of Pleistocene hominin evolution.”* While we recognize that this may indeed be a commonly held opinion in our field, it is not a correct one. Nevertheless, we have gone to great pains to explain this position more thoroughly throughout the manuscript. We hope this (and other explanatory additions throughout) serves to clarify this matter with regard to what we *do* know about the phylogenetic position of *H. naledi* and what we do not, being fair to the poor nature of the rest of the comparative hominin record, and that this satisfies these reviewers’ concerns for this popularly held belief.

Concerning Reviewer #2’s comment, *“The authors make the case that* Homo naledi *could be from early* Homo *stock in the Pliocene, based on the analyses by Dembo et al.* But *those analyses are heavily influenced by the small brain size of the original naledi finds. How do the authors expect this will change with the larger brain size estimates? What does this mean for the phylogenetic hypotheses they offer?”* In fact these (and other) studies were not heavily influenced by brain size estimates. Nevertheless, we have included (as discussed above) a more detailed explanation of our interpretations of the phylogenetic origins of *H. naledi* and hopes this clarifies this matter.

Concerning Reviewer #2’s comment *“Could the authors please provide some discussion about morphological similarities between the Florisbad molar (likely associated with the skull) and that of Homo naledi? Not just a size comparison. Is it possible these are same species, or is the morphology different?”* We have not conducted this study, as much of the specimen was destroyed during ESR dating and additionally, as Reviewer #3 points out, this is likely associated with a hominin cranium of very different morphology to *H. naledi* (a comparison we did conduct in Berger et al. 2015). We also feel such specialized anatomical studies are perhaps better suited to more detailed upcoming studies on the dentition of *H. naledi* rather than in these papers.

[Further changes were requested before acceptance.]

Deposition: We have considered very carefully the reviewer and editorial comments concerning mortuary behavior by *H. naledi*. We have adopted precisely the form suggested by the editors and reviewers: The manuscript now presents non-hominin agency as a null hypothesis, and discusses the evidence from both chambers as it relates to that hypothesis. We have added clear examples of non-hominin depositional processes and the evidence for them in other complex cave situations with multiple deposits of hominin fossil material. This discussion now ends with a brief statement that the hypothesis of mortuary behavior is not equivalent to symbolic behavior, and that in our assessment it was likely within the capabilities of other species of *Homo*.

This treatment of the issue has increased the length of this section of the paper, which the editorial and reviewer comments also suggested, but which we had hesitated to do in earlier versions of the manuscript. But we think that it is much clearer now how the new information from the Lesedi Chamber affects the interpretation, and we are also explicit about the ways that the new findings do not affect the interpretation.

To be clear, we do not intend for this synthesis of many aspects of the *H. naledi* record to present new evidence concerning the deposition of the fossil remains. We can see that for the editors and reviewers this has been a major topic of discussion. If the revisions are still not suitable, we would be willing to remove the section and continue forward without it though we feel the submission would be the poorer without it.

Gene flow: We have added a paragraph to the section on mosaic anatomy as suggested, which provides further discussion of the anatomy of *H. naledi* in light of adaptive and non-adaptive evolutionary processes, following on the conclusions of Laird et al. (2017).

Implications section: We have addressed each of the detailed comments listed and have created a table (Table 1) and a Figure (Figure 1) that lists fossils discussed and their dates as well as the location of relevant sites as it seems that reviewers were having difficulty in a simple written summary of the fossils and their dates. We hope these improve the readability of the manuscript.

We have added missing references and corrected the other minor editorial suggestions as listed.